# Cefotaxime Exposure-Caused Oxidative Stress, Intestinal Damage and Gut Microbial Disruption in *Artemia sinica*

**DOI:** 10.3390/microorganisms12040675

**Published:** 2024-03-28

**Authors:** Huizhong Pang, Kaixuan Zheng, Wenbo Wang, Mingjuan Zheng, Yudan Liu, Hong Yin, Daochuan Zhang

**Affiliations:** 1The International Centre for Precision Environmental Health and Governance, College of Life Sciences, Hebei University, Baoding 071002, China; huizhongpang@hotmail.com (H.P.); zhengkaixuan1207@hotmail.com (K.Z.); wangwenbo02@hotmail.com (W.W.); zhengmingjuan02@hotmail.com (M.Z.);; 2Key Laboratory of Zoological Systematics and Application of Hebei Province, College of Life Sciences, Hebei University, Baoding 071002, China

**Keywords:** cefotaxime, oxidative stress, intestinal tissue, gut microbiota, *Artemia sinica*

## Abstract

Cefotaxime (CTX) is an easily detectable antibiotic pollutant in the water environment, but little is known about its toxic effects on aquatic invertebrates, especially on the intestine. Here, we determined the oxidative stress conditions of *A. sinica* under CTX exposure with five concentrations (0, 0.001, 0.01, 0.1 and 1 mg/L) for 14 days. After that, we focused on changes in intestinal tissue morphology and gut microbiota in *A. sinica* caused by CTX exposure at 0.01 mg/L. We found malondialdehyde (MDA) was elevated in CTX treatment groups, suggesting the obvious antibiotic-induced oxidative stress. We also found CTX exposure at 0.01 mg/L decreased the villus height and muscularis thickness in gut tissue. The 16S rRNA gene analysis indicated that CTX exposure reshaped the gut microbiota diversity and community composition. Proteobacteria, Actinobacteriota and Bacteroidota were the most widely represented phyla in *A. sinica* gut. The exposure to CTX led to the absence of Verrucomicrobia in dominant phyla and an increase in Bacteroidota abundance. At the genus level, eleven genera with an abundance greater than 0.1% exhibited statistically significant differences among groups. Furthermore, changes in gut microbiota composition were accompanied by modifications in gut microbiota functions, with an up-regulation in amino acid and drug metabolism functions and a down-regulation in xenobiotic biodegradation and lipid metabolism-related functions under CTX exposure. Overall, our study enhances our understanding of the intestinal damage and microbiota disorder caused by the cefotaxime pollutant in aquatic invertebrates, which would provide guidance for healthy aquaculture.

## 1. Introduction

*Artemia sinica* is a tiny halophilic invertebrate living in saline waters. These zooplankton are widely used to feed fish and shrimp larvae in aquaculture, and artificial feed cannot completely replace the use of *Artemia* due to its high nutrition [1]. *Artemia* can also serve as a drug carrier owing to its non-selective filtration. In turn, pollutants in water environment can easily accumulate in its body and passed to downstream of food chain [2,3]. These characteristics make them occupy an important trophic level in aquatic food chain. Moreover, the short life cycle of *Artemia* enable them suitable as laboratory animal with high value for application in toxicity study [4]. So far, the toxicity of various substances has been tested on *Artemia* including heavy metals, nanoparticles, sewages, and pharmaceuticals. Most assessments focused on hatching, mortality, and swimming for detecting acute/chronic toxicity [5,6,7]. Meanwhile, Next Generation Sequences (NGS) approaches provide a new light to evaluate the toxicity reaction of *Artemia* from micro level [8,9].

Antibiotics have been considered important environmental pollutants due to their overuse in human therapy and other commercial activities, including livestock and aquaculture [10,11]. The presence of antibiotics in natural ecosystems has a huge impact on organisms. Previous studies have supported the idea that multiple antibiotics can generate harmful reactive oxygen species (ROS) regardless of drug–target interactions, and widely induce oxidative stress [12]. Antibiotic-induced oxidative stress as a toxic effect has been reported in animals and plants, which would interfere with their survival and development, leading to inhibition in growth and reproduction, physiological, biochemical and tissue alterations [13,14,15,16]. In addition, the impact of antibiotics on the microbiosphere is of great concern [17]. Low-level antibiotic exposure could change the diversity and composition of environmental microbial communities significantly [18]. Moreover, altered microbial community structure caused by antibiotic pollution would also affect bacterial enzyme activity and lead to an increase in parasites and pathogens [11].

Cephalosporin is a commonly used antibiotic for the treatment and prevention of bacterial diseases, for example, genital tract infections in humans, respiratory tract infections and intramammary infections in animals [19]. The concentration of cephalosporins detected in different water samples varies from ng/L to mg/L [20]. Cefotaxime sodium (CTX), which belongs to the third-generation cephalosporin, is the third most-detected cephalosporin in water environments [19]. In China, β-lactam antibiotics represented by CTX were used as a substitute for sulfonamides [21]. In eastern China, the detection frequency of CTX was over 40% in fish ponds around Taihu Lake in summer with the antibiotic concentration exceeding 10,000 ng/L [22]. CTX was also the most abundant antibiotic in surface water of Xiangjiang River in central China with a maximum concentration at 830 ng/L [23]. Tang detected cephalosporins at 0.974 mg/L and CTX at 0.183 mg/L in pharmaceutical wastewater, and the concentrations remained as high as 0.703 mg/L and 0.108 mg/L, respectively, after anaerobic-anoxic-aerobic (A^2^O) treatment [24]. In Hong Kong and Shenzhen sewage samples, CTX reached concentrations exceeding 1 μg/mL [25]. In addition, integrated fish-livestock farming is a typical breeding system in southern China, but the large amount of antibiotics used in livestock breeding will aggravate the pollution of the water environment [26]. He [27] and Zhou [28] used CTX at concentrations ranging from 0.1 to 3 mg/L to simulate antibiotic contamination of water bodies in an integrated farming system. Recent studies have reported the ecotoxicity of CTX on aquatic vertebrates using zebrafish as a model [29]. CTX stress decreased the body size, affected swimming behavior with changes in gene expression related to the nervous system and sensory organs [30,31], and disrupted the oxidative balance of zebrafish [32]. Furthermore, CTX in aquatic surroundings enhanced the β-lactam resistance and its transmission mobility in zebrafish bodies, which suggests a potential risk of antibiotic-resistant pathogens spreading from environmental ecosystems to clinical patients, posing a threat to human health [33]. Dutta [34] studied the median effect concentration (EC50) of CTX on *Daphnia magna*, but the mechanism of the CTX effect on aquatic invertebrates is still unclear.

As one of the most complex and abundant ecosystems in the host, the gut microbiota intervenes in the production of ROS and impacts the intestinal mucosal barrier [35]. How antibiotics affect hosts through gut microbes has been studied in mammalian models [36,37]. Antibiotic pollutants such as tetracyclines and sulfanilamide influence the gut of aquatic organisms as well [38], but little is known about the impact of cephalosporin remaining in the water environment. To investigate the imbalanced oxidation-reduction status and intestinal alteration caused by CTX stress, we performed cefotaxime exposure on pseudo-adults of *A. sinica* for 14 days. We will (1) detect the oxidative stress caused by antibiotics via the total antioxidant capacity (T-AOC) and MDA content; (2) observe the effect of antibiotic stress on *A. sinica* morphology and gut tissue through stereo microscopes and hematoxylin-eosin staining; and (3) study the changes in the gut microbiome using full-length 16S rRNA gene sequencing. Our aim is to explore the toxicity of cefotaxime on the gut system in aquatic invertebrates using *A. sinica* as a model. It would help with the healthy aquaculture and provide a scientific basis for ecological risk assessment of cephalosporin contaminants.

## 2. Material and Methods

### 2.1. Animal Culture, Solution Prepare and Antibiotic Exposure

Cysts of *A. sinica* were collected from the Salt Lake of Yuncheng, Shanxi, China, in August 2020. The cysts were hatched and maintained in sterilized artificial seawater (28 °C, 28‰ salinity) under a 12: 12 LD light cycle (1000 Lux light intensity). After one day of incubation, the larvae were fed with dry yeast (Angel, Inner Mongolia, China) once a day. The culture condition was set following Wang [39], which is the optimal environment for *A. sinica* in the laboratory.

Cefotaxime (CAS-64485-93-4) was purchased from Aladdin Scientific (Shanghai, China) with purity ≥ 98%. The standard solution (10 mg/L) was prepared each time before use by dissolving 1 mg of the reagent in 100 mL of sterilized artificial seawater with 28‰ salinity. The experimental solution was prepared by diluting a standard solution with sterilized artificial seawater.

In order to exclude the influence of gender, antibiotic treatments were applied to the pseudo-adult stage of *A. sinica* with no differentiation of genital organs after a seven-day culture. The pseudo-adult *A. sinica* cultured in sterilized artificial seawater without CTX exposure was used as the control group (0 mg/L CTX). The treatment concentrations for CTX exposure were set at 0.001 mg/L, 0.01 mg/L, 0.1 mg/L, and 1 mg/L, respectively. For each concentration, three replicates with 30 individuals each were conducted. The experimental conditions were same as those used in the cultural period. The antibiotic exposure experiment lasted 14 days.

### 2.2. Oxidative Stress Assessment

The *A. sinica* individuals of each concentration were washed three times with 0.01 M phosphate-buffered saline (PBS) at 4 °C. Then, the samples were dried, weighed, and put into a tissue grinder immediately with 0.01 M PBS at a ratio of 1:9 for crushing. After centrifuging the homogenate at 2500 rpm for 10 min, the supernatant was collected for biochemical parameter measurements. The T-AOC and MDA contents were analyzed using commercial kits (Nanjing Jiancheng Bioengineering Institute, Nanjing, China) according to the manufacturer’s instructions. The T-AOC and MDA concentrations were expressed as U/mg protein and nmol/mg protein, respectively. T-AOC and MAD were measured three times for each concentration. Differences between the control group and the treatment groups were compared by means of one-way ANOVA and Dunnett’s test.

### 2.3. Morphology and Gut Histomorphology

Subsequent experiments were performed on the control group and the 0.01 mg/L CTX treatment group based on the results of oxidative stress quantification. The abbreviations “CK group” and “CTX group” were used to represent the control group and the 0.01 mg/L CTX treatment group. Individuals from the CK group and CTX group were checked for their morphology using a Leica M205A to confirm whether antibiotic exposure changed their appearance. After that, *A. sinica* specimens from two groups (*n* = 6, three specimens per group) were fixed with 4% paraformaldehyde for 24 h, followed by dehydration in ethanol with concentrations at 75%, 85%, 90% and 100%, and then transferred to xylene for transparency processing, and embedded in paraffin wax. The midgut of each specimen was sliced transversely into 4 μm sections and stained using hematoxylin-eosin (H&E). The sections were observed under the Olympus CX21FS1. The villi height (V) and muscularis thickness (M) were measured with ten replicates from each slide. Differences between the CK group and the CTX group were compared by means of Welch’s *t*-test.

### 2.4. DNA Extraction and PacBio Sequencing of 16S rRNA Gene

The head, thorax, and appendages of *A. sinica* were removed after washing with PBS, and we collected the remaining abdomen for further gut microbial DNA extraction. Fifteen individuals from each group were gathered into the same 1.5 mL EP tube to form a sample (0.03 g per sample) with three replicates for each concentration. The total DNA was extracted with the TGuide S96 Magnetic Soil/Stool DNA Kit (Tiangen Biotech Co., Ltd., Beijing, China) according to the manufacturer’s instructions. The DNA quantity and quality were measured with a Nanodrop 2000 Spectrophotometer (Thermo, Waltham, MA, USA).

The whole region of the 16S rRNA gene was amplified with a universal primer pair (27F-AGRGTTTGATYNTGGCTCAG, 1492R-TASGGHTACCTTGTTASGACTT). After a purification process, the products were sequenced on the PacBio platform by Biomarker Technologies Corporation (Beijing, China). The optimized circular consensus sequences (CCS) were obtained after filtering with a threshold of minPasses ≥ 5, minPredictedAccuracy ≥ 0.9, and length between 1200 and 1650 bp. The UCHIME algorithm was used in detecting and removing chimera sequences to obtain clean reads.

### 2.5. Bioinformatics and Statistics

Operational taxonomic units (OTUs) were generated by clustering sequences at the 97% similarity level on QIIME2. Statistics on community composition in each sample were calculated at the level of phylum, class, order, family, genus, and species based on the SILVA database. Alpha diversity was determined based on the ACE, Chao1, Simpson and Shannon indexes using Wilcoxon tests. Principal Component Analysis (PCA) and permutational multivariate analysis of variance (PERMANOVA) with Adonis based on the weighted-unifrac distance metric were used to analyze the beta diversity. Metastats and Lefse analysis were performed to identify differentially abundant taxa. Furthermore, PICRUSt2 was employed to predict the functional potential of the microbial communities. The significance of the difference in function abundance between groups was evaluated in STAMP using Welch’s t test. The significance level was set at a *p*-value of *p* < 0.05.

## 3. Result

### 3.1. Effect of CTX on Oxidative Stress in A. sinica

The activity of T-AOC and the MDA content were measured in *A. sinica* after the fourteen-day CTX treatment. As shown in Figure 1, there was no significant difference between the 0.001 mg/L CTX treatment group and the control group in T-AOC level (*p*-value > 0.05). However, the T-AOC level increased greatly in the 0.01, 0.1 and 1 mg/L CTX treatment groups (*p*-value < 0.05). The content of MDA increased significantly in all treatment groups (*p*-value < 0.05) compared to the control group. The peak of MDA content was 1.49 nmol/mg, with the highest T-AOC activity at 0.01 U/mg under 0.01 mg/L CTX exposure.

Since the maximum level of T-AOC and MDA occurred under 0.01 mg/L CTX exposure, we compare the differences between the control group and the 0.01 mg/L CTX treatment group on morphology, intestine histology, and gut microbiota. The abbreviations “CK group” and “CTX group” were used to represent the control group and the 0.01 mg/L CTX treatment group, respectively, in the subsequent results.

### 3.2. Morphology Changes and Destruction of Intestinal Structural Integrity

Compared with the CK group, individuals in the CTX group showed weak extensional and uneven appendages, which is the hallmark of unhealthy *Artemia*. We also observed several black spots on the back of *A. sinica* in the CTX group (Figure 2a). Besides the changes in the external morphology of *A. sinica*, cefotaxime exposure altered the intestinal structure (Figure 2b). The intestinal mucosa of the CK group had an intact structure, and the intestinal villi were dense and evenly distributed with clear boundaries. In the CTX group, the intestine of *A. sinica* was swelling with unclear intercellular junctions in intestinal tissues, and the intestinal villi were sparse and varied in length. Villus height and muscularis thickness were extremely decreased in the CTX group (*p*-value < 0.001) compared to the CK group after fourteen days of treatment (Figure 3a,b). The results of H&E staining suggested that the intestinal structure was damaged by cefotaxime treatment.

### 3.3. Gut Microbial Community Composition and Diversity

A total of 42,063 CCS sequences obtained from six samples (three samples per group) by barcode-based identification were sorted into 261 operational taxonomic units (OTUs). The taxonomy classification was identified as 13 phyla, 21 classes, 63 orders, 106 families, 176 genera and 221 species. The Venn diagram illustrated two groups sharing 108 OTUs, 84 genera and 10 phyla (Appendix A). The CK group owned a unique bacteria phyla Spirochaetota, while the CTX group had two unique bacteria phyla, Dependentiae and Patescibacteria.

Bacterial groups with a relative abundance of ≥5% are defined as dominant at the phylum level. Proteobacteria was the most widely represented phylum in two groups, with a relative abundance of 37.48% in the CK group and 44.57% in the CTX group, respectively. Actinobacteriota accounted for 32.27% and 10.50%, respectively. The exposure to CTX increased the proportion of Bacteroidota, with a relative abundance of 5.29% in the CK group and 21.64% in the CTX group, respectively. Moreover, the relative abundance of Verrucomicrobia in the CK group was 9.54%, but below 5% in the CTX group (Figure 4a). At the genus level (Figure 4b), the most abundant genera in the CK group were *Gordonia* (27.00%), *GKS98_freshwater_group* (18.37%) and *unclassified_Bacteria* (7.68%). The genus *TM7a* (10.81%) was the predominate genus in the CTX group, followed by *Marinobacter* (7.11%), *Labrenzia* (6.78%) and *Methylophaga* (5.78%).

The α-diversity was reflected by the ACE, Chao1, Simpson and Shannon indexes (Appendix A). No significant effect was observed on α-diversity in gut samples (*p*-value > 0.05) between the two groups. Nevertheless, the PCA plot (Appendix A) showed that the samples of the CK and CTX groups were clustered into two different groups. In addition, the PERMANOVA result (R^2^ = 0.514, *p*-value = 0.001) indicated significant microbial community differences between the two groups.

### 3.4. Differences in the Gut Microbiome between Groups

The Metastats results showed that only the abundance of Bacteroidota increased significantly (*p*-value < 0.05) in the CTX group at the phylum level. At the genus level, eleven genera with an abundance greater than 0.1% showed significant differences (*p*-value < 0.05) relative to the control (Figure 5). Among the eight genera with increased abundance, four (*Ahrensia*, *Chromatocurvus*, *Pseudomonas* and *Roseovarius*) belong to Proteobacteria, and three (*Algoriphagus*, *Bizionia* and *Psychroflexus*) belong to Bacteroidota. Among the three genera with decreased abundance, both *GKS98 _freshwater_group* and *Marinicella* belong to Proteobacteria.

The cladogram from phylum to species with an LDA score > 3 was constructed to identify the taxa contributing to the differences between the two groups (Figure 6). Sixteen biomarkers were found in the CK group, including the phylum Planctomycetota, order Izemoplasmatales, Gammaproteobacteria, family DEV007, Amoebophilaceae, and species *Legionella londiniensis*. The CTX group was enriched in phylum Bacteroidota, Patescibacteria, order Rhodobacterales, Saccharimonadales, family Marinobacteraceae, Halieaceae, Beijerinckiaceae, genus *Vallitalea* and *Ahrensia*.

### 3.5. Function Prediction of Gut Microbiota

Forty-three potential functions in total at the class 2 level were predicted based on the KEGG database (Figure 7). The main functional prediction categories were related to metabolism, including amino acid (8.08%, 7.73%, respectively), carbohydrate (8.91%, 8.57%, respectively), and energy (4.33%, 4.27%, respectively) for two groups of *A. sinica*. Furthermore, we analyzed the functional differences between groups at the class 3 level of the KEGG database (Figure 8). Most of the predicted functions with differential abundance were concentrated in metabolic pathways. The exposure of CTX up-regulated the functions related to streptomycin biosynthesis, amino acid (cysteine and methionine, lysine biosynthesis) metabolism and drug metabolism. Conversely, xenobiotic biodegradation (toluene, aminobenzoate, ethylbenzene and atrazine degradation) functions and lipid metabolism-related functions, including the PPAR signaling pathway, peroxisome, synthesis and degradation of ketone bodies, were down-regulated after the exposure to 0.01 mg/L CTX. In addition, we also found that the abundance of functions related to the FoxO signaling pathway decreased in the CTX group.

## 4. Discussion

Clinically relevant levels of antibiotics can induce reactive oxygen species (ROS) overproduction and mitochondrial dysfunction in mammalian cells, leading to oxidative stress and oxidative damage [40]. As one of the cellular targets of ROS, oxidative stress can induce lipid peroxidation, which leads to the generation of a variety of oxidized products [41]. MDA is the final decomposition product of lipid peroxidation, which has been commonly used as a biomarker to measure the level of oxidative stress and the degree of cellular damage [42,43]. In this study, the significant increase in MDA in all CTX treatment groups indicated that antibiotic exposure in the water environment can induce oxidative stress in *A. sinica*. We speculate that the antioxidant systems of *A. sinica* were activated correspondingly to protect the body from the destruction of ROS [44], as shown by the elevation of T-AOC in CTX treatment groups. However, although *A. sinica* enhanced its antioxidant capacity to resist the toxic effects of oxidative stress, it did not prevent the aggravation of lipid peroxidation in the body. In addition, it is noteworthy that the MDA levels in the 0.1 and 1 mg/L treatment groups decreased compared to the 0.01 treatment group. Besides being excreted directly in the feces [45], MDA can be reduced to non-toxic alcohols by aldehyde dehydrogenase (ALDH) [46]. A similar decreased MDA response was reported in some aquatic organisms, which could be related to the activation of ALDH by xenobiotic exposure [47,48,49]. Enhanced ALDH metabolism in the body could eliminate excess MDA, but Gariac considered that MDA itself does not appear to directly activate ALDH [49]. What exactly causes ALDH activation remains to be further studied.

The oxidative stress in the gut contributes to developing intestinal pathologies, including damage to the intestinal mucosal barrier [50,51]. Intestinal villus height and muscularis thickness can be used to quantify the degree of intestinal damage. Compared to the CK group, the intestinal villi of *A. sinica* atrophied, the mucus muscle layer became thinner, and the epithelial structure was lost in the CTX group. The damage to intestinal tissue, with villi structure lost and villi height shortened, was also observed in cephalosporin-treated neonatal mice [52]. The effect of antibiotic exposure on the intestine at an early life stage could last into adulthood. After stopping antibiotic treatment in neonatal mice for 35 days, the damaged intestinal epithelium recovered partly, but the body weight was still significantly lower than that of the control group [53]. Moreover, the reduced muscularis thickness leads to an increase in gut permeability, which makes it vulnerable to external attacks [54]. The intestinal pathological changes in this study indicated that cefotaxime exposure has impaired the physical barrier of the intestinal mucosa. The integrity of the intestinal mucosa ensures its complete function to maintain homeostasis and regulate the immune response [55]. A compromised intestinal barrier can lead to the over-activation of the gut immune system, inducing systemic inflammation or an impaired immune response [56]. This might be related to the black dots we observed on *A. sinica* in the CTX group, which are melanin depositions resulting from inflammatory responses [57].

Oral cephalosporin treatment has been proven to induce gut microbiota dysbiosis [58]. We characterized the effect of cefotaxime in the water environment on the gut microbiome. Although the α-diversity showed no significant differences, the intestinal microbial community composition was distinct between the CK and CTX groups. Proteobacteria, Actinobacteria and Bacteroidota are dominant components of the gut microbiota in Crustacea [59,60,61], and they have been reported as the main phyla in *Artemia* sp. through DGGE [62]. We found a new dominant phylum, Verrucomicrobiota, in the CK group. Lefse results also showed that the CK group had a higher relative abundance of the Verrucomicrobial family DEV007. The bacterial phylum Verrucomicrobia is common in soil, freshwater, marine and hypersaline systems and has the capacity for polysaccharide degradation [63]. It is also a dominant microbial community in the intestines of some aquatic animals [64,65,66,67]. Verrucomicrobia in the human intestine have been identified as mucin degraders and considered a hallmark of glucose homeostasis and a healthy gut [68]. The abundance of Verrucomicrobia decreased in the diseased *Litopenaeus vannamei*, which was speculated to be related to the weakened feeding activity [69]. The absence of Verrucomicrobiota in the dominant phyla of the CTX group could be a result of oxidative stress caused by antibiotic exposure, which can affect the balance of the gut microbiota. Meanwhile, deregulated bacteria can produce more ROS and further exacerbate oxidative stress [70].

Treatment of CTX increased the abundance of Bacteroidota in the samples, which was consistent with the changes in gut microbiota under β-lactam antibiotic therapy [71,72]. Bacteria of this phylum can enhance the metabolic efficiency of macromolecular organic matter, i.e., proteins and polysaccharides [73]. This could account for the increased abundance of amino acid metabolism function in the CTX group. However, propionate, acetate, and succinate produced by Bacteroidota can reduce the assembly of tight junctions and increase intestinal permeability [74]. The increased abundance of Bacteroides has been detected in chronic inflammation or metabolic diseases associated with intestinal barrier damage [75,76,77]. We considered that the damage to gut tissue in *A. sinica* was closely related to the elevation in Bacteroidota. For marine ecosystems, the existence of Bacteroidota causes devastating and widespread disease outbreaks in eukaryotic hosts, with an abundance of diseased individuals [78]. The Lefse results showed that Flavobacteriales and Cytophagales were significantly higher in the CTX group. These two types of bacteria are widely reported and characterized as pathogens. Flavobacterial disease is recognized as a serious threat to fish framing, and Flavobacterial infections were also reported in shrimp, crabs, amphibians, and humans [79,80,81]. Kayani [82] found that the low antibiotic exposure to zebrafish in the early life stage led to a large accumulation of pathogenic Flavobacterial species in the gut, which may predispose zebrafish to health-related complications. Cytophagales are facultative predators that can extract nutrients by preying on other intestinal bacteria. The increased abundance of Cytophagales in the gut was correlated with inflammation markers [83,84]. In addition, *Cytophaga* spp. have been identified as pathogens for various algal diseases [78]. Disease and mortality caused by Bacteroidota infection have been challenges for the aquaculture industry. The enrichment of these two potentially pathogenic Bacteroidetes in the CTX group, Flavobacteriales and Cytophagales, should be given more attention.

Although the abundance of Proteobacteria did not vary at the phylum level, most bacterial species with significant differences belong to Proteobacteria. The enriched Proteobacteria populations in the CTX group included *Ahrensia*, *Chromatocurvus*, *Pseudomonas*, and *Roseovarius* at the genus level, and Marinobacteraceae, Halieaceae, Rhodobacteraceae, Xanthobacteraceae, and Beijerinckiaceae at the family level. Proteobacteria are considered a signature of dysbiosis in the gut microbiota [85]. The disruption feature of gut microbiota in the antibiotic-induced grass carp model is a significant expansion of Proteobacteria [86]. The CTX exposure in water may disrupt the anaerobic environment of the gut [87] in *A. sinica*, which leads to the overabundance of Proteobacteria. Interestingly, two potential pathogenic genera of Proteobacteria exhibited opposite abundance changes: the *GKS98_freshwater_group* declined in the CTX group, whereas *Pseudomonas* raised significantly relative to the control. In nature, the abundance of these two genera is affected by salinity, nitrogen content, and other environmental factors [88,89,90]. *Pseudomonas* species can invade multiple organ systems in humans and other animals. An increased abundance of pathogenic bacteria may have a toxicological effect on *Artemia*. Administration of cefotaxime via water or injection is considered effective for *Pseudomonas* infection in aquaculture [91]. However, an increase in the abundance of *Pseudomonas* was observed in wastewater, livestock manure, and patients after antibiotic treatment [92,93,94]. *Pseudomonas* is the main potential host for multiple antibiotic resistance genes (ARGs), and Kang [93] speculated that the *Pseudomonas* biofilm reduces the ability of antibiotics to kill bacteria, which would play a role in antibiotic resistance. The enrichment of this genus is significantly correlated with the high abundance of ARGs [95,96]. The increase in *Pseudomonas* in the *A. sinica* gut may be attributed to the expansion of antibiotic-resistant bacteria under CTX stress. On this foundation, further studies on the ARGs of *A. sinica* after antibiotic exposure are needed to elucidate the impact of antibiotic resistance on aquatic organisms. Our results also suggest that it is necessary to consider the health risks associated with pathogen hazards due to cefotaxime pollution in water environments.

The present study demonstrated the oxidative stress caused by cefotaxime contamination in aquatic invertebrates and its damage to the intestine. We also found a new dominant phylum, Verrucomicrobia, in the *A. sinica* intestine and revealed the disorder of the gut microbiota after cefotaxime exposure. The changes in gut microbiota structure subsequently affected the essential metabolic function of the intestine, thereby leading to potential health hazards.

## Figures and Tables

**Figure 1 microorganisms-12-00675-f001:**
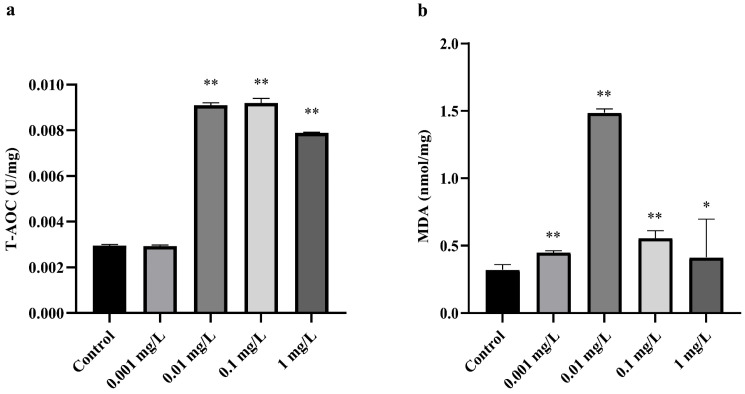
The effects of CTX exposure at five different concentrations for 14 days on (**a**) T-AOC and (**b**) MDA in *A. sinica*. * and ** represent significant differences between the control group and the CTX treatment group. * *p*-value < 0.05, and ** *p*-value < 0.01. T-AOC = total antioxidant capacity, MDA = malondialdehyde.

**Figure 2 microorganisms-12-00675-f002:**
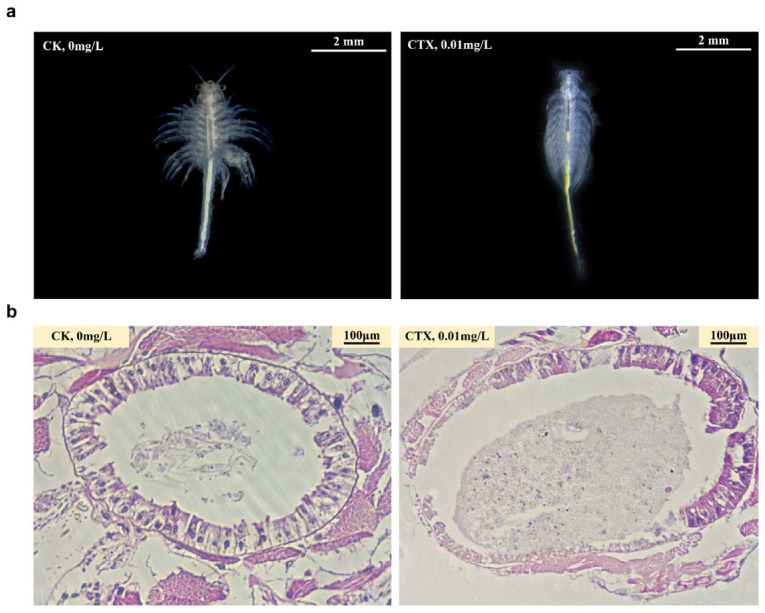
Morphological changes in the appearance (**a**) and intestinal tissues (**b**) of *A. sinica* under 0.01 mg/L CTX exposure. CK = the control group, CTX = the 0.01 mg/L CTX treatment group.

**Figure 3 microorganisms-12-00675-f003:**
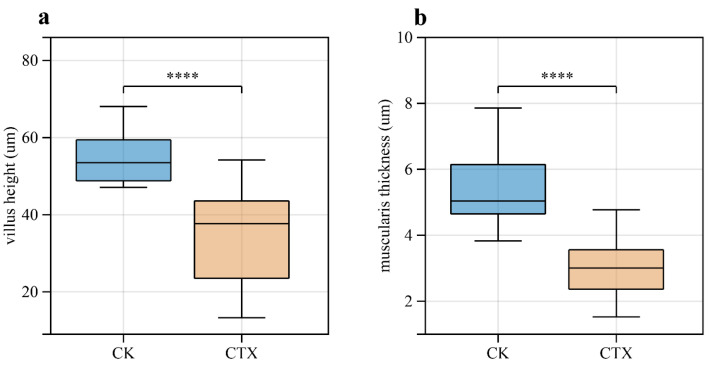
Effects of 0.01 mg/L CTX on intestinal structure of *A. sinica* in (**a**) villus height; and (**b**) muscularis thickness. **** represents significant differences (*p*-value < 0.001) between the CK group and the CTX group. CK = the control group, CTX = the 0.01 mg/L CTX treatment group.

**Figure 4 microorganisms-12-00675-f004:**
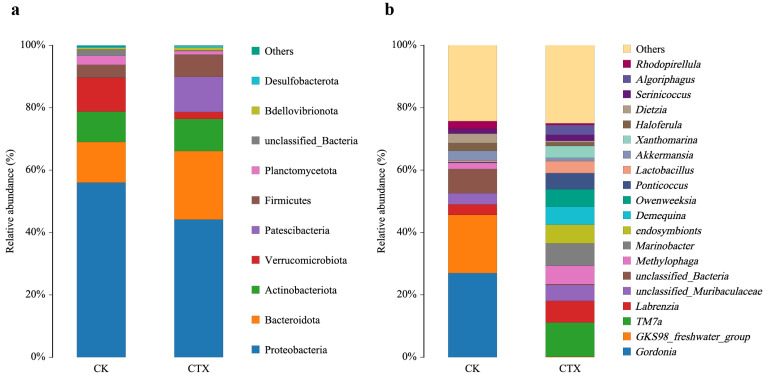
Relative abundance of (**a**) top 10 bacterial at phylum level; and (**b**) top 20 bacterial at genus level in *A. sinica* gut microbiota of the CK and CTX groups. CK = the control group, CTX = the 0.01 mg/L CTX treatment group.

**Figure 5 microorganisms-12-00675-f005:**
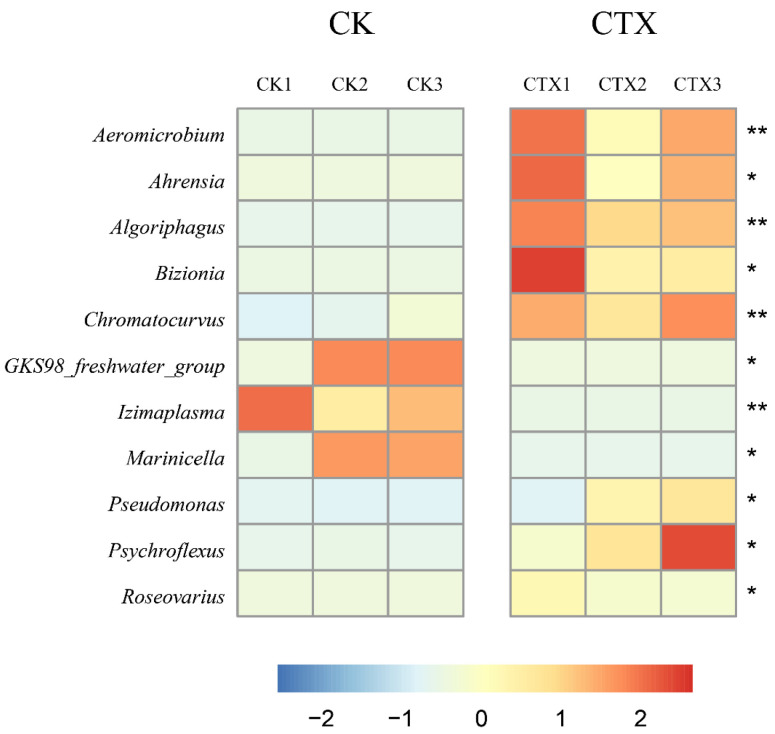
Genera with significant differences in abundance of gut microbiota in *A. sinica* after 0.01 mg/L CTX exposure. * and ** represent significant differences between the CK group and the CTX group (* *p*-value < 0.05, ** *p*-value < 0.01). The values shown on heatmap were z-scores generated by z-normalization of relative species abundance. CK1, CK2 and CK3 were the three samples from the CK group, while CTX1, CTX2, CTX3 were the three samples from the CTX group. CK = the control group, CTX = the 0.01 mg/L CTX treatment group.

**Figure 6 microorganisms-12-00675-f006:**
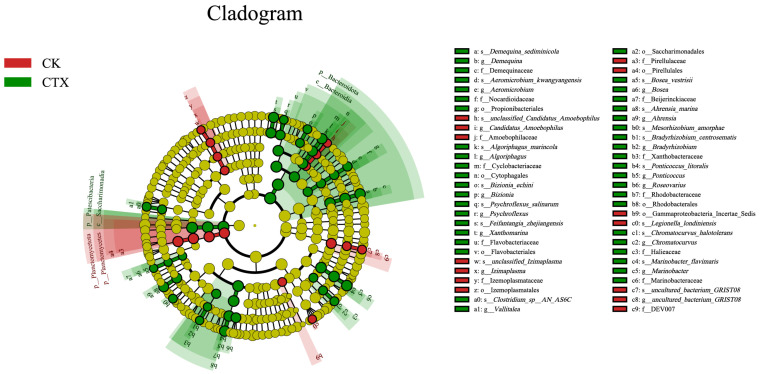
The cladogram to depict the key and most differentially abundant taxa associated with CTX exposure in *A. sinica* gut microbiota. Logarithmic LDA score = 3, and *p*-value = 0.05. CK = the control group, CTX = the 0.01 mg/L CTX treatment group.

**Figure 7 microorganisms-12-00675-f007:**
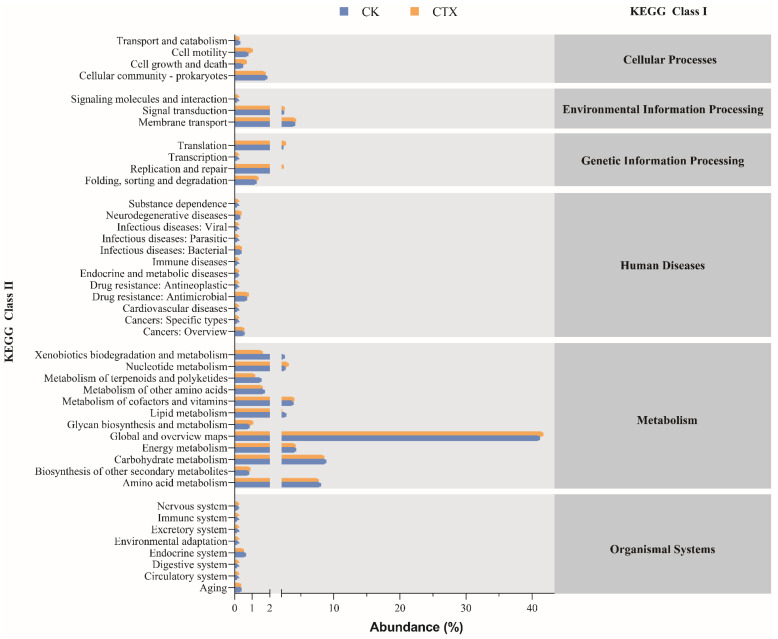
Mean relative abundance of predicted functions at KEGG class 2 level in *A. sinica* gut microbiota of the CK and CTX group. CK = the control group, CTX = the 0.01 mg/L CTX treatment group.

**Figure 8 microorganisms-12-00675-f008:**
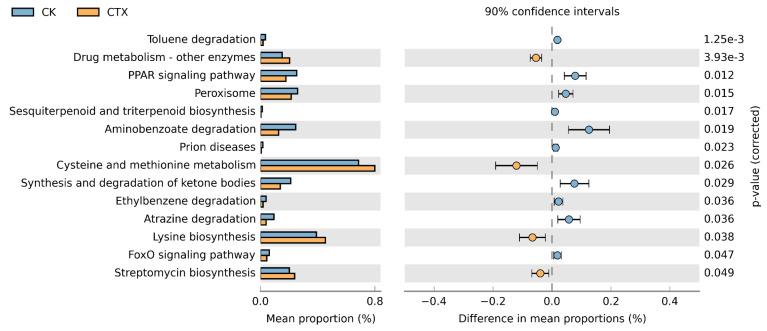
Functional differences at KEGG class 3 level between the CK and CTX groups. CK = the control group, CTX = the 0.01 mg/L CTX treatment group.

## Data Availability

16S rRNA-seq data were deposited into the NCBI BioProject database (PRJNA998203).

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
