# Peer review of "Cefotaxime Exposure-Caused Oxidative Stress, Intestinal Damage and Gut Microbial Disruption in Artemia sinica"

_microorganisms, 2024, doi:10.3390/microorganisms12040675_

Round 1

Reviewer 1 Report

Comments and Suggestions for Authors

In this manuscript Panga et al. report on alarming problem of environmental pollution from drugs. Particularly, the significant alteration to the Artemia sinica intestinal microbiota is highlighted. A topic is addressed that should be explored further, as it is useful for improving the well-being of these invertebrate and global health of living beings and environment.

Overall, the approach taken is scientifically sound. The studies were performed at the proper methodological level and with rigor; the results are substantiated and well discussed with a logical conclusion.

I only noticed a few small flaws in the exhibition form. Below, there are some suggestions that are aimed at improving the quality of the submitted manuscript.

1.     Taxonomic note: the name of Artemia sinica and the other species mentioned in the manuscript must always be written in italics.

2.     Figures: for greater clarity, I recommend adding the meaning of the acronyms (CTX, CK and PCA,….) in the captions.

3.     Be careful to always specify the meaning of the acronyms when using it for the first time (for example MDA is never indicated in full).

4.     A comment about the MDA levels reduction in samples treated at higher CTX concentration is useful and adequately justify the exclusion of samples from subsequent analyses.

5.     If available, data on the effects of CTX on the survival of A. sinica larval stages would also be interesting.

Author Response

Dear reviewer, 

We feel great thanks for your professional review work on our article. The point by point response to your comments is listed in the pdf file. Please see the attachment. We hope the correction will meet with approval.

Sincerely,

Daochuan Zhang

Reviewer 2 Report

Comments and Suggestions for Authors

Experimentally,  the paper is sound in the science done, and its  data interpretation.  However,  since the antibiotic concentration used are not those found in natural waters (levels would be much lower), the interest value of this paper is much reduced.  The concern is the effects seen in the Artemia are due to the toxicity of the unrealistically high antibiotic concentration used.  Even in mammals, it is well understood that using high doses of antibiotics could be toxic to the organism, and especially its gut microbiota. That this is seen in Artemia is therefore not at all surprising. The relevance of the antibiotic exposure at levels found within the environment needs to be demonstrated.

Comments on the Quality of English Language

Do another proof read concentrating on sentence construction.

Author Response

(The authors gave the same response as above.)

Round 2

Reviewer 2 Report

Comments and Suggestions for Authors

Thank you to the authors for their amendments, which have improved the clarity of the manuscript.  However, while the intro does now cite info on antibiotic levels in several environmental sites, the levels used in the study are still much higher than the new text refers to. It is known that high levels of most antibiotics are toxic, so the challenge for the authors is showing the relevance of the doses they use. That is, can the doses used in the study that had an effect on the Artemia be found within the environment?

Author Response

Dear revierwer, 

Thank you for your valueable feedback. Please see our response in the attachment. 

Sincerly,

Daochuan Zhang

Round 3

Reviewer 2 Report

Comments and Suggestions for Authors

I still have a few reservations about the biological validity of the antibiotics used, but the authors do (mostly) acknowledge the experimental nature of their findings.